# Mapping Roads in the Brazilian Amazon with Artificial Intelligence and Sentinel-2

**Jonas Botelho, Jr.** , **Stefany C. P. Costa** , **Júlia G. Ribeiro and Carlos M. Souza, Jr.** *

IMAZON-Amazon Institute of People and the Environment, Belém 66055-200, PA, Brazil;
jonas@imazon.org.br (J.B.J.); stefany.pinheiro@imazon.org.br (S.C.P.C.); juliagabriela@imazon.org.br (J.G.R.)
* Correspondence: souzajr@imazon.org.br; Tel.: +55-91-3182-4000

**Abstract:** This study presents our efforts to automate the detection of unofficial roads (herein, roads) in the Brazilian Amazon using artificial intelligence (AI). In this region, roads are built by loggers, goldminers, and unauthorized land settlements from existing official roads, expanding over pristine forests and leading to new deforestation and fire hotspots. Previous research used visual interpretation, hand digitization, and vector editing techniques to create a thorough Amazon Road Dataset (ARD) from Landsat imagery. The ARD allowed assessment of the road dynamics and impacts on deforestation, landscape fragmentation, and fires and supported several scientific and societal applications. This research used the existing ARD to train and model a modified U-Net algorithm to detect rural roads in the Brazilian Amazon using Sentinel-2 imagery from 2020 in the Azure Planetary Computer platform. Moreover, we implemented a post-AI detection protocol to connect and vectorize the U-Net road detected to create a new ARD. We estimated the recall and precision accuracy using an independent ARD dataset, obtaining 65% and 71%, respectively. Visual interpretation of the road detected with the AI algorithm suggests that the accuracy is underestimated. The reference dataset does not include all roads that the AI algorithm can detect in the Sentinel-2 imagery. We found an astonishing footprint of roads in the Brazilian Legal Amazon, with 3.46 million km of roads mapped in 2020. Most roads are in private lands (~55%) and 25% are in open public lands under land grabbing pressure. The roads are also expanding over forested areas with 41% cut or within 10 km from the roads, leaving 59% of the 3.1 million km$^2$ of the remaining original forest roadless. Our AI and post-AI models fully automated road detection in rural areas of the Brazilian Amazon, making it possible to operationalize road monitoring. We are using the AI road map to understand better rural roads' impact on new deforestation, fires, and landscape fragmentation and to support societal and policy applications for forest conservation and regional planning.

**Keywords:** artificial intelligence; Amazon; road extraction; deep learning; U-Net; Sentinel-2; planetary computer

## 1. Introduction

Mapping roads from satellite imagery is an important image processing research topic to improve urban planning, transportation systems, and agricultural organization [1,2]. For tropical forest regions, such as the Amazon biome, roads are one of the main drivers of forest change by deforestation [3–5], increasing the likelihood of fires [6] and threats to protected areas [7]. Therefore, mapping and monitoring roads from space are crucial for identifying threats to tropical forests and the tradition and indigenous people living in the region. The first efforts to map roads in the Brazilian Amazon biome used visual interpretation of Landsat imagery [8,9], revealing an overwhelming spread and extent of illegal roads in this region. However, mapping and monitoring roads were still dependent on visual interpretation mapping protocols, which are time-consuming [8,10] and prone to biases of human performance [11].

It is also difficult to automate the detection and mapping of roads using medium-spatial-resolution satellite data such as Sentinel-2 (i.e., 10–20 m pixel size). The road detection algorithm has to account for geometric (length, width, and shape), radiometric (spectral response), topological (connectivity), functional (use), and textural (local spectral variability) attributes to build a fully automated road detection model [12]. Because of the challenges described above to automate road detection and mapping, researchers had used Landsat satellite imagery to develop a road mapping protocol for the Brazilian Amazon biome based on visual interpretation followed by hand digitizing [6,7]. The road mapping protocol was applied to map and monitor road expansion over several years (2008 through 2016), resulting in an unprecedented Amazon Road Dataset (ARD). The ARD has been used to understand road geometry and spatial pattern and its correlation with deforestation and forest fragmentation [13,14], the proximity effect of roads to deforestation [7] and fires [6], and to improve deforestation risk models [15,16]. Additionally, the ARD has also been used to elucidate road-building processes, functions, and drivers [17,18] and to understand the impact of roads on biodiversity [19]. These ARD applications demonstrate the relevance of road mapping and monitoring to scientific, conservation, and policy applications. However, manual mapping of roads with satellite imagery is a laborious task, making frequent monitoring over large areas challenging.

Automated road detection methods have been developed for very-high-spatial-resolution imagery based on deep learning algorithms [12,20]. Artificial Intelligence (AI), particularly the convolutional neural network (CNN) deep learning algorithm, has been used successfully to detect and map rural roads in large forested areas of Canada with RapidEye imagery [11]. The AI-CNN method in Canada did not produce full-connected vectorized roads, requiring post-classification techniques, resulting in a recall accuracy of 89–97% and precision of 85–91% [11]. A U-Net [21], a CNN variant, was first used for segmentation in medical imaging. It is [22] considered a state-of-art for image segmentation because it is based on the deconstruction and reconstruction of images for feature extraction, increasing object detection in various applications [2,23,24]. U-Net's differentiating characteristics are its accuracy and high-speed discriminative learning capacity from a small number of training images [21]. As a result, several U-Net remote sensing applications have been proposed, including road extraction using high-resolution imagery [2,22–24]. Therefore, the U-Net algorithm is promising to automate the detection of rural roads of the Brazilian Amazon, overcoming the visual interpretation mapping protocol [8] used to build the ARD. We modified the original U-Net algorithm, which erodes image input chips (256 × 256 pixels), making road discontinuity a problem, and changed the activation and loss functions of the model to make it more sensitive to detect roads using Sentinel-2 10 m spatial-resolution imagery.

This study then used the ARD to train our modified U-Net algorithm to detect rural roads in the Brazilian Amazon. First, we selected a sub-area of the ARD to randomly define samples to train, calibrate, and test the U-Net model to detect roads using Sentinel-2 imagery. Then, we applied it to the U-Net road model to map roads over the entire Brazilian Amazon. Next, we implemented a post-AI road detection algorithm to generate a fully connected vectorized road map for 2020. Furthermore, we built a workflow to integrate the Sentinel-2 OpenHub and Microsoft Azure Cloud Platform to implement the U-Net road and post-AI road detection algorithms. Sentinel OpenHub was used to select and filter cloud-free Sentinel-2 scenes. At the same time, Azure provided the computational power to preprocess the Sentinel-2 imagery, train and test our modified U-Net model, and export the data to our Azure Blob Storage. Our main goal with this study is to develop an automated AI model to monitor roads more frequently in the Brazilian Amazon and assess the correlation of forests with roads. Our road detection models applied to the Sentinel-2 images allowed mapping of the road location and extension over the entire Brazilian Legal Amazon, showing that an intricate and complex road network cuts a significant percentage of the Brazilian Amazon forest. The AI road model revealed more roads than

human-based road mapping efforts using moderate-spatial-resolution imagery, improving the understanding of their harmful ecological effects.

## 2. Materials and Methods

We used Microsoft Azure Planetary Computer resources to acquire and store data from Sentinel-2 and ARD to train and make predictions with our modified U-Net road detection model. We selected the Sentinel-2 satellite imagery based on location, date, bands, and cloud cover. Then, we used the ARD to define random sample locations (i.e, image chips of 256 × 256 pixels) for acquiring training data for the AI road detection algorithm, composed of sampled Sentinel-2 images and rasterized binary images of roads. Finally, we converted the gathered information into the TFRecord format to minimize size and seemingly integrate with the Tensorflow AI library inside the Microsoft Azure Cloud Machine Learning resource. We implemented a post-AI processing workflow to remove classification errors, automate road vectorization, and connect isolated road segments. Figure 1 summarizes the entire U-Net road detection and post-AI algorithms. We provide information about the study area and explain the steps of the road detection algorithms in detail in the following sections.

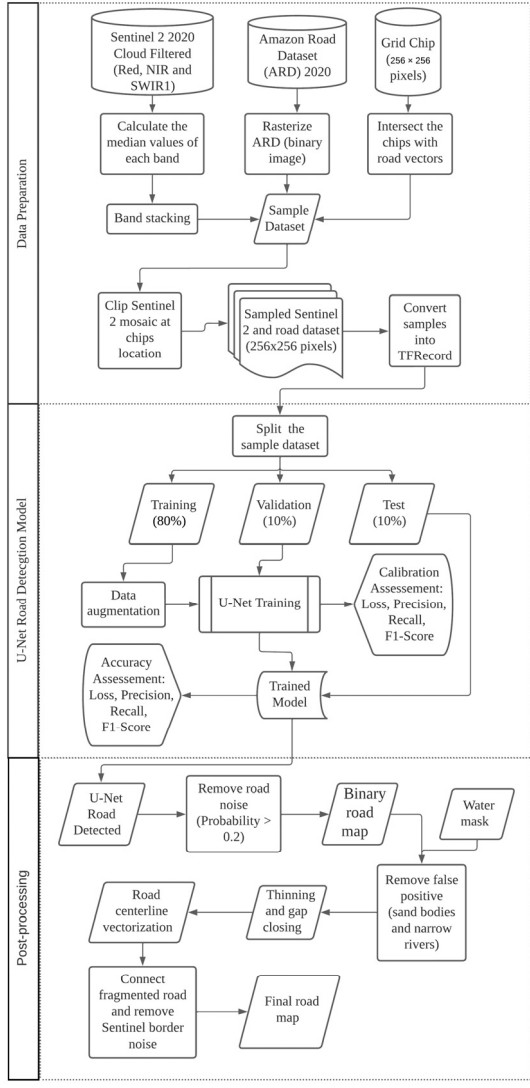

**Figure 1.** Workflow of the U-Net road detection and post-processing applied to Sentinel-2 imagery.

### 2.1. Study Area

The study area covers the entire Brazilian Legal Amazon area. The training chips' location concentrated on the Southern Region of the Pará state, which has diverse road geometries (i.e., dendritic, geometric, and fishbone) [25] (Figure 2). The road geometry diversity increased the variety in our training dataset by providing different road shapes and forms for the AI model to generalize the presence of rural roads in the Amazon region. The sub-area chosen for training and validating the model is approximately 586 thousand km$^2$ and is delimited by eight International Millionth Map of the World sheets (SB-21-X, SB-21-Y, SB-21-Z, SB-22-V, SC-21-V, SC-21-X, and SC-22-X) [26] (Figure 2). The sub-area is mainly covered by dense forests, including old-growth and secondary forests [27], and by pasturelands, logging, and, to a lesser extent, gold mining. We then sampled the region into small chunks of data (chips) and randomly selected 2500 chips based on road density estimated from the ARD. The samples were composed of chips with 256 × 256 pixels of Sentinel-2 images at 10 m spatial resolution collected from June 2020 to October 2020.

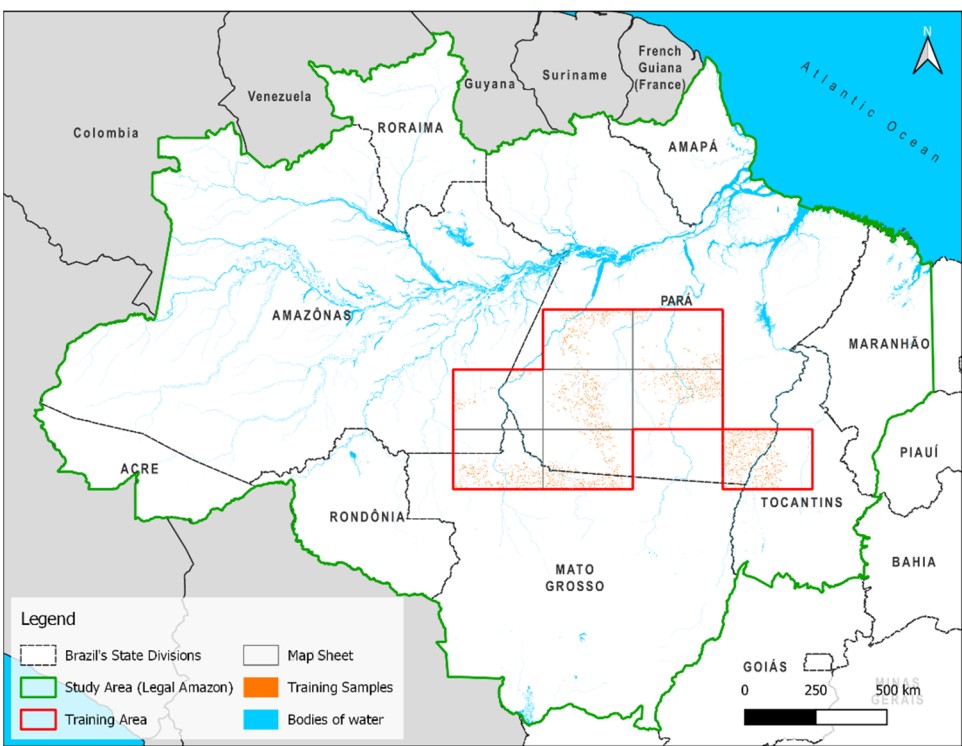

**Figure 2.** The study area covers the entire Brazilian Amazon (green). Training and validation chips (orange) were selected in eight International Millionth Map of the World map sheets (red; SB-21-X, SB-21-Y, SB-21-Z, SB-22-V, SC-21-V, SC-21-X, and SC-22-X).

### 2.2. Satellite Data to Detect Roads

We used Sentinel-2 imagery data to train and apply the AI road detection model at a 10 m pixel size. We acquired the Sentinel-2 image scenes using the Azure Blob data request API by filters including scene location, date, bands, and percent of cloud cover. Acquired during June and October 2020, the image scenes included the Shortwave Infrared 1 (SWIR1), Near-Infrared (NIR), and Red spectral bands with less than 30% cloud cover. We chose the timespan for image acquisition between June and October due to a lesser cloud persistence. In addition, the bands SWIR1, NIR, and Red are more suitable for detecting roads in non-urban areas and differentiating from other linear features (e.g., powerlines and geological lineaments) [8].

With the selected Sentinel-2 scenes, we then built a spatial-temporal mosaic by calculating the median of each band separately and stacking them afterward. The final preprocessing step included the application of histogram contrast stretching to enhance

the types of roads we aimed to detect with the AI U-Net algorithm (Figure 3). Image analysts have recognized three types of roads using RGB color composite images from the Amazon [8,9]: (i) visible: continuous straight or curved lines visible to the naked eye; (ii) fragmented roadways: discontinued straight or curved lines that are not continuous but discernible to the naked and possibly traced and connected; (iii) partially visible roads: linear characteristics, straight or curved, directly visible in the color composite but recognized and digitized based on their context and spatial arrangement (i.e., adjacent deforested areas and canopy damage due to selective logging) (Figure 3).

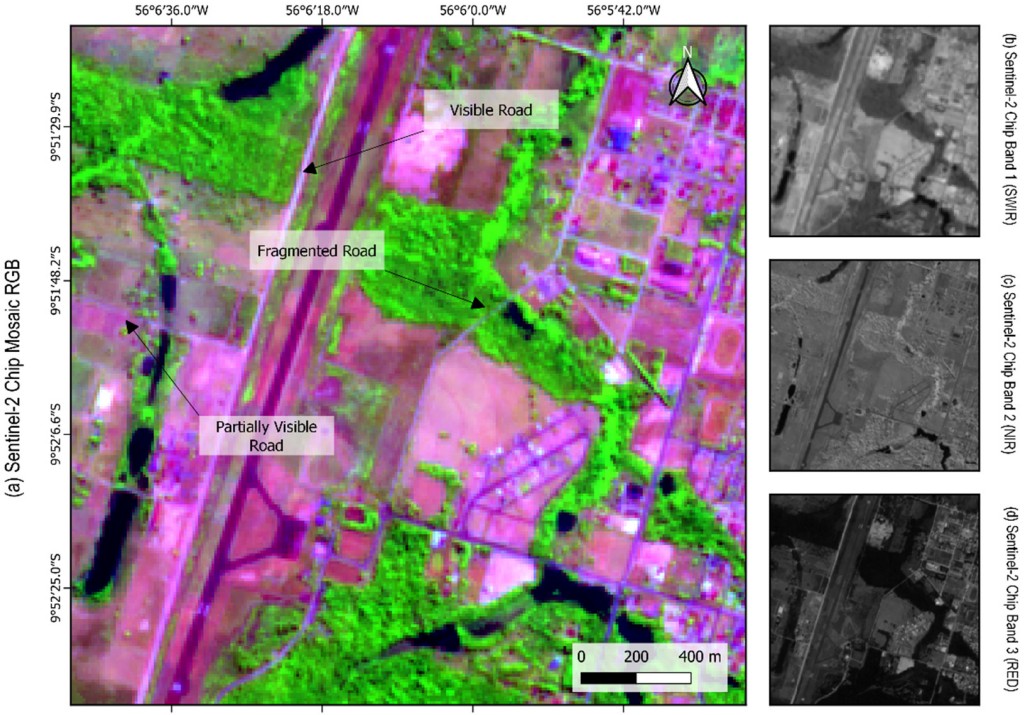

**Figure 3.** Example of Sentinel-2 RGB (SWIR, NIR, and Red) median mosaicking used to train the U-Net model to detect roads in the Amazon biome. The color contrast stretched image was used to build the training dataset through visual interpretation and hand digitizing three types of roads: visible, fragmented, and partially visible, based on Ref. [8].

### 2.3. Amazon Road Dataset (ARD)

The ARD encompasses ten years of road mapping through visual interpretation and manually traced mapping roads using a protocol for mapping unofficial roads with satellite imagery in the Amazon [8]. This road dataset is composed of official roads (80 thousand km) and unofficial roads (454 thousand km) totaling 534 thousand kilometers. The ARD mapping protocol was applied to different satellites throughout the years, including Landsat-5 (2006–2011), Resourcesat (2012), and Landsat-8 (2016). We used the ARD 2016 data to sample areas of interest to build our training, calibration and test datasets. We applied a vector grid to divide the region into smaller chunks of data over the study area. These sample points were selected randomly based on the road's density (i.e., km of roads per $km^2$) inside the chip areas of $256 \times 256$ pixels—approximately 6.5 $km^2$.

We randomly sampled 2500 chips to manually remap roads to update the road map to Sentinel-2 2020 imagery. This process created new road information for the chip areas at a 10 m spatial resolution to train, calibrate, and validate the U-Net road detection model. We sampled the chip areas (6.5 $km^2$) for updating the road information for the AI model into five categories:

- No road;
- 0 < road density ≤ 0.15 km;

- 0.15 < road density ≤ 0.76 km;
- 0.76 < road density ≤ 1.53 km;
- Road density > 1.53 km.

The number of image chips samples per road density is shown in Figure 4. We observed an increase in samples with road density. Therefore, we manually added roads to the Sentinel-2 2020 samples, following Ref. [8] to update the 2016 ARD dataset (Figure 4).

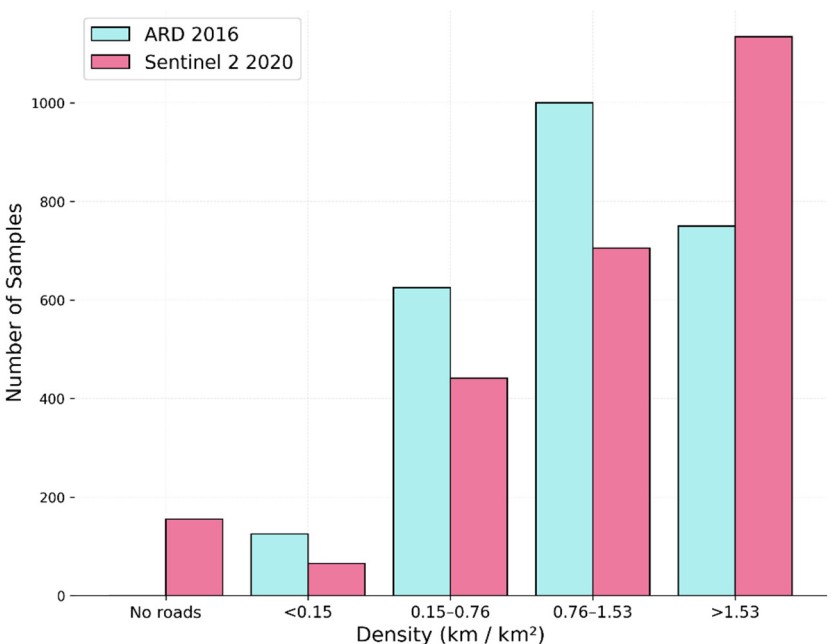

**Figure 4.** The number of chip samples per road density and a comparison between ARD 2016 (Landsat based, 30 m pixel) and 2020 (Sentinel-2, 10 m pixel) manually mapped for building the training dataset.

### 2.4. Sampling and Data Preparation

To maximize training speed and performance, we broke down the Sentinel-2 mosaic and the ARD raster into smaller chunks of data by creating a grid of 256 × 256 cells within the training area. We used the existing ARD from 2016 to select grids that intersected with the ARD data to build a reference road dataset to train the U-Net road detection model. Next, the Sentinel-2 image was clipped with the selected grids (256 × 256 grid), resulting in 2500 samples. We updated the roads within the sampled grids through visual interpretation and hand digitizing following the protocol proposed by Ref. [8], which was further rasterized to a 10 m pixel size. Finally, we stacked the Sentinel-2 RGB bands and the updated raster roads and divided the 2500 chip samples into training, validation, and test datasets. We randomly split the chip samples in 80% for training purposes and separated the 20% equally for validation and test, i.e., 250 each (Figure 1).

Before exporting the datasets to Azure Blob Storage, we applied data conversion from raster data into TFRecord tensor arrays compatible with Tensorflow data API [28], which facilitated data consumption during training. In addition, we added each chip's coordinates information to the TFRecord, consolidating its contents: image data information (pixel values), upper left x-coordinate, upper left y-coordinate, lower right x-coordinate, and lower right y-coordinate (Figure 1).

To minimize the effects of our class imbalance and to prevent overfitting our model, we applied data augmentation processes to the datasets to increase the variability number of samples. The transformations applied to the images were rotations of 90°, 180°, and 270° degrees; horizontal and vertical mirroring; three repetitions, increasing the number of tiles from 2500 to 23,400.

### 2.5. U-Net Model for Road Detection

We modified the original U-Net architecture [21] layers and hyperparameters to detect roads from Sentinel-2 imagery. First, regarding layers, we altered the padding parameters on convolutional layers to create a zero-padding border around the images, allowing for the same input (256 × 256) and output (256 × 256) data size to prevent border data neglection [29,30]. Furthermore, instead of using the Rectified Linear Unit activation function (ReLU), we opted for its variation, Leaky Rectified Linear Unit (LeakyReLU) [31], due to the dying neuron problem. In addition, we added a dropout layer of 0.3 before the last convolutional layer to prevent overfitting [32], which provided variability to the model structure. Finally, the soft dice loss function [33] replaced the original loss function (i.e., pixel-wise cross-entropy loss function) to minimize the impact of imbalanced classes for image segmentation problems and Nadam [34] as the optimizer (Figure 5).

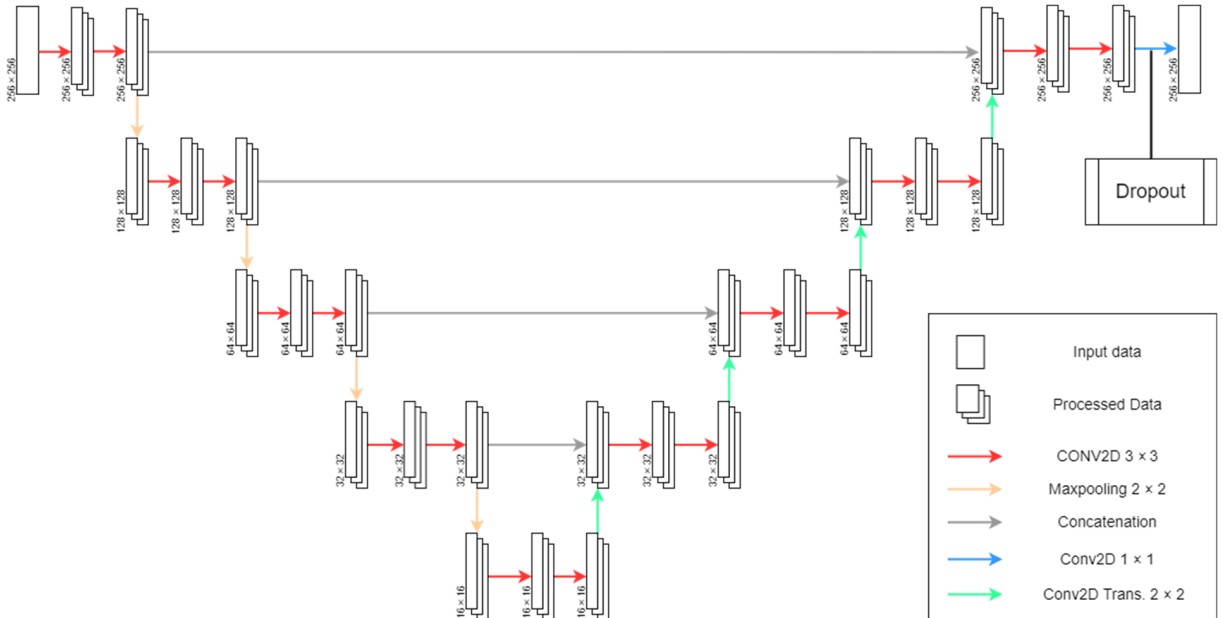

**Figure 5.** The modified U-Net model for road detection and mapping applied to 10 m pixel-size Sentinel-2 imagery based on Refs. [21,30].

As for the model training parameters, we applied an 8-size mini-batch with a learning rate of $10^{-4}$ set to iterate over 25 epochs. We set the learning process to stop at the 25th epoch because the loss value was minimized around the 21st and 23rd. We also included other training parameters to prevent unnecessary computational power usage and save checkpoint versions of the model, such as early stopping and TensorFlow checkpoints. We set early stopping to end the training process if there was no improvement over nine epochs, which helped with the generalization problem and prevented resource usage. We also defined checkpoints to assess whether the validation loss function value had improved between epochs. The implementation code of the U-Net road detection model can be accessed on Github through https://github.com/JonasImazon/Road_detection_model.git (accessed on 1 June 2022).

### 2.6. Accuracy Assessment

We performed the prediction accuracy analysis of our U-Net road model using the validation and test datasets (250 chip samples each). Our validation sample metrics (precision, recall, and *F1-Score*) were calculated during training and saved onto a log file to assess the model calibration. In addition, we used an independent test dataset to generate the following accuracy metrics: user's, producer's, and *F1-Score* (i.e., a harmonic mean calculated from both precision and recall values) metrics on both datasets. The accuracy

metrics assessed are shown below. First, the *Precision* metric utilizes *TP* (true positives) and *FP* (false positives) to assess the number of actual valid positive results over the model's total number of positive cases.

$$Precision: \frac{TP}{TP + FP} \tag{1}$$

Differently, *Recall* uses *FN* (false negatives) to measure the number of true positives over the reference data total number of positive information:

$$Recall: \frac{TP}{TP + FN} \tag{2}$$

The culmination of both *Precision* and *Recall* lies in the usage of *F1-Score*.

$$F1\text{-}Score = 2 \times \frac{Precision \times Recall}{Precision + Recall} = \frac{2 \times TP}{2 \times TP + FP + FN} \tag{3}$$

We calculated the accuracy assessment metrics in a pixel-wise approach, allowing us to estimate for each pixel *TP*, *TN* (true negative), *FP*, and *FN* (false negative).

### 2.7. Post-Processing

Given that the U-Net road model outputs the same data type (i.e., a raster image) as its inputs, we ended with an image representing each pixel's probability of being one of two classes: road and no-road. To detect the road center line, we first empirically define a confidence threshold of 20% (i.e., pixels with a probability value equal to or greater than 0.2) over the output values, creating a road binary image. The output image data (in raster format) are helpful for visual interpretation and comparison between the road mapping results and the reference road dataset. However, the road raster output does not provide road attribute information regarding length, shape, and connectivity. We then applied post-processing techniques to convert the raster road map into a vector road center line and extracted attributes of road segments (e.g., length and connectivity). These processes include georeferencing the TFRecord file, road vectorization, connecting flawed road segments, and noise removal (e.g., natural geomorphological lineaments such as narrow rivers and geological faults). We explain these steps in detail below.

First, we applied the georeferencing process by accessing each chip's coordinate information presented in its TFRecord file. Then, using Gdal [35], we set the result image's geographic location by setting its extent and resolution (10 m). Each georeferenced image is stored in memory to be quickly accessed by the following processes. Next, we applied a water body mask with a 200 m buffer to remove false positives created by natural linear features (i.e., primarily channels) along rivers. Furthermore, a sequence of dilation and erosion processes was applied to fix some of the missing links between road segments using a geometric orientation of a 10 × 1 rectangle. The rectangle was rotated around each pixel to increase the reach between close pixels. Finally, a skeletonization process was applied to create 1 × 1 pixel segments.

Finally, we converted the chip's raster data into vector data using the Grass Python Module [36]. In addition, we used the model also to remove other identified false positives, such as small unconnected segments and scene borders. We divided the process into four steps: (i) deletion of segments with a length less than 1 km; (ii) removal of segments that match the geographic location of charts borders; (iii) removal of segments that match the geographic location of scenes border (i.e., 100 m); (iv) deletion of segments inside areas with road density less than 10 km per 100 km$^2$. These steps allowed us to deal with different categories of false positive road detection in an automated manner. The final road map is in a shapefile format containing segments of roads in vector format, which enables statistical and geospatial analyses and also the cartographic representation of the road dataset, including estimation of the total length.

## 3. Results

### 3.1. U-Net Road Model

We implemented a modified U-Net for road detection in the Azure platform with the Machine Learning Studio instance configured with one NVIDIA Tesla K80, six cores, and 56 GB RAM. The total run time to train the proposed neural network was approximately 2 h. We chose the soft dice loss function results as a checker for the calibration and selection of the U-Net road model as it is the model's convergence indicator. Furthermore, the soft dice loss function is ideal for imbalanced datasets [22], which minimizes the results to obtain a maximum overlap between reference and predicted data. Figure 6 shows that our model started the training process with a validation loss value of ~0.8, indicating poor performance. However, as the training progressed along the epochs, we observed that the difference in the loss values between validation and training decreased, indicating convergence and stabilization of the model.

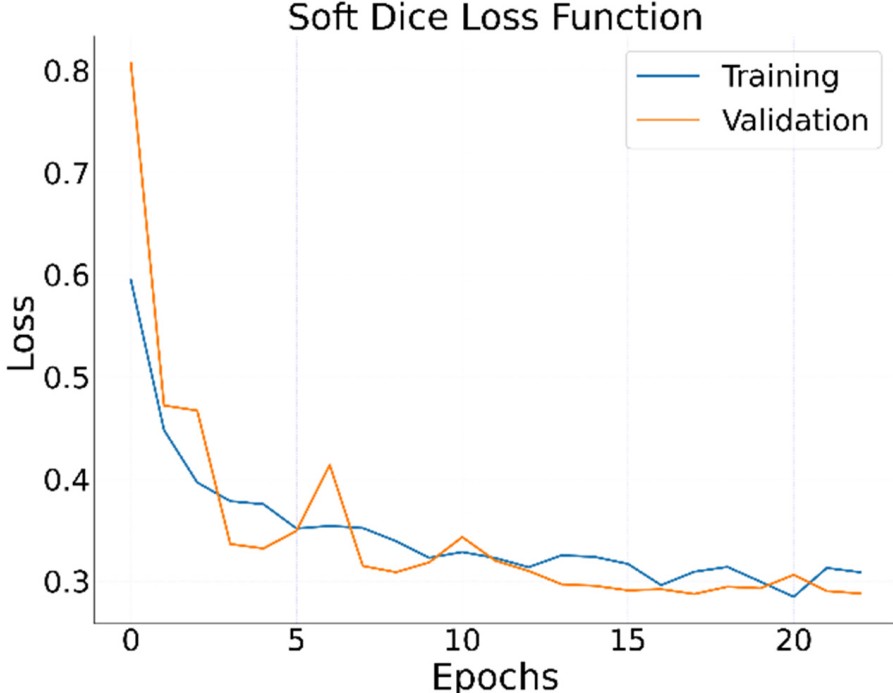

**Figure 6.** Model's training and validation Soft Dice Loss curve along 25 epochs.

We capped epoch iterations at the 25th epoch due to improvement stagnation. In addition, by visually analyzing the results, we noticed that our model reached the generalization point and satisfactory road detection after the 15th epoch, not improving significantly after it. We ordered the loss function values from max to min to verify the best iteration of our model. The best epochs reached similar results: 16th (0.292), 15th (0.291), 21st (0.290), 22nd (0.288), and 17th (0.287) (Figure 6). We then chose the lowest loss function result, presented by the 17th epoch, based on the minimization of the loss function. However, we did not test each of the mentioned iterations for visual checking.

### 3.2. Road Mapping

We applied the trained model to our entire dataset (training, validation, and testing) ($\approx$16,250 km$^2$) to check its visual accuracy. The prediction process took approximately 49 s to generate 2500 binary images. Although the U-Net road model showed connectivity flaws and background noise, the model predicted most of the roads presented in these datasets. Those false positives were mainly found close to rivers and open areas with strip-tillage where their spectral characteristic of the images was similar to roads'. In addition, the model predicted other roads that were not included in the reference data, indicating the

presence of human error while composing our reference dataset (which potentially affected our accuracy assessment analysis, as we further discuss) [37].

We selected examples of road types to demonstrate the generalization capacity of our U-Net road model (Figure 7). By visually assessing the road mapping results relative to the Sentinel-2 input image, we can infer the following conclusions about the model:

1.  The model can detect roads similar to the reference data, leaving some road segments disconnected.
2.  Roads detected from the Sentinel-2 imagery are wider because adjacent pixels (i.e., 1 to 3 pixels wider) to roads are also detected.
3.  The U-Net road model might clump closed road segments as individual roads in dense road areas.
4.  The model detected more 'partially visible roads' than roads mapped visually to generate the reference dataset, especially in private rural lands.
5.  We also found false-positive roads (not new roads) created near water bodies' borders and on the edge of forest areas, and in agriculture fields (i.e., strip tillage).

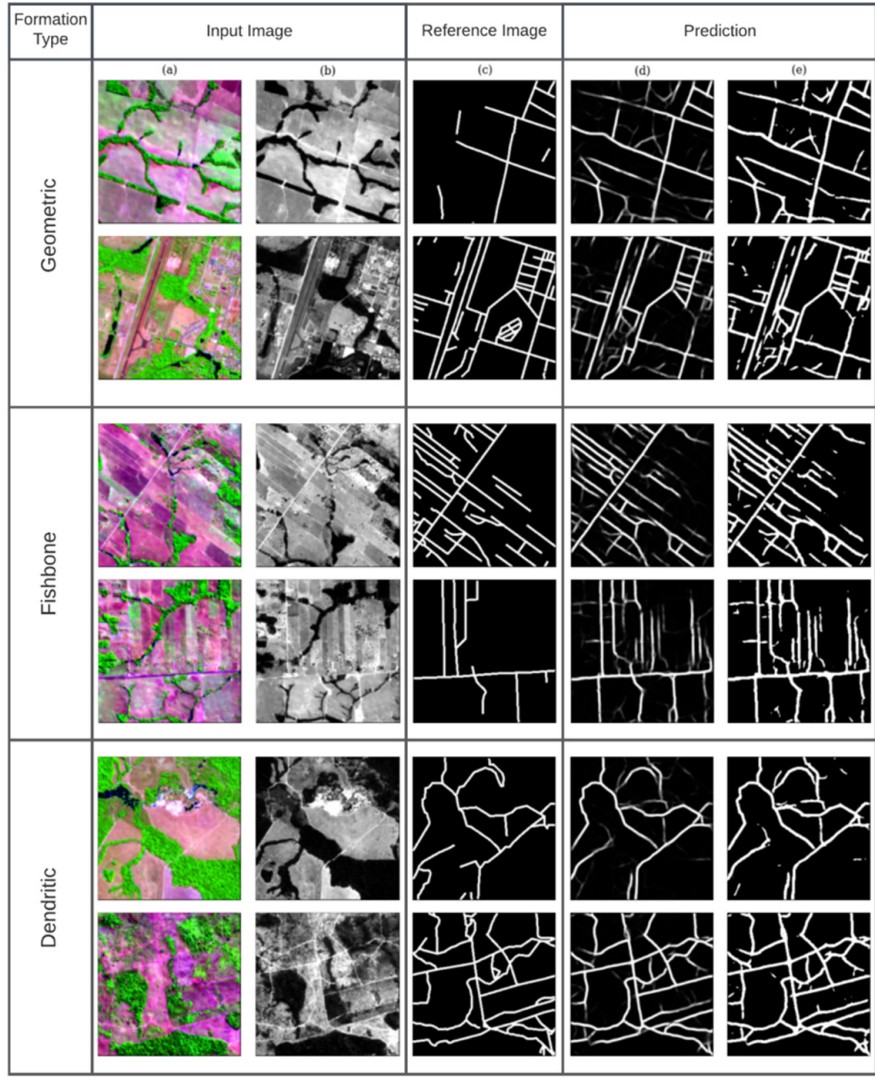

**Figure 7.** Comparison between the reference data and the predicted roads with the U-Net model. (**a**) RGB (SWIR1, NIR, and Red) Sentinel-2 input image for road prediction. (**b**) Grey level Sentinel-2 Red band for better visualization of roads. (**c**) Manually mapped reference data. (**d**) Raw road prediction result. (**e**) Filtered result with more than 20% confidence used as input for the post-classification.

We also assessed the road U-Net model's accuracy using the user's (prediction) and producer's accuracy (recall) and the *F1-Score* metrics. First, we calculated these metrics using the training dataset through each epoch of the training process. During the training phase, our model reached a 69% precision, 64% recall, and 68% *F1-Score*. Then, we calculated the accuracy metrics using the validation and test datasets (i.e., not used to train the U-Net road model), achieving a user's accuracy of 72.2% and 71.7%, respectively. The producer's accuracy (61.1% and 65.6%) and *F1-Score* (65.5% and 68.4%) were lower than the user's accuracy for validation and test datasets (Figure 8).

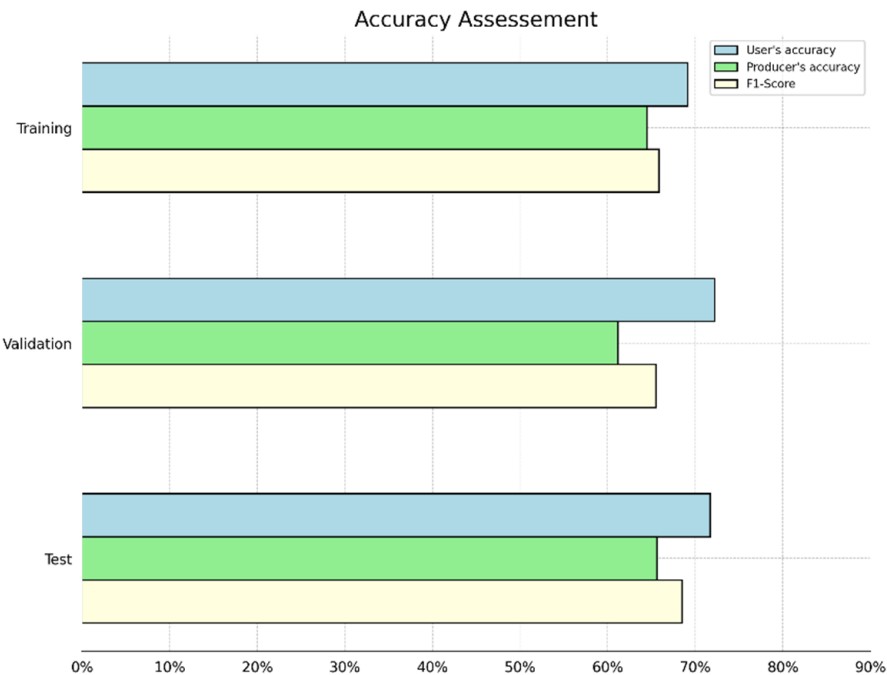

**Figure 8.** Accuracy of the U-Net road model for the training, validation, and test datasets.

As previously mentioned, these accuracy metrics may underestimate the accuracy of the U-Net road model. A careful visual comparison of the reference data against the roads detected with the U-Net revealed unmapped roads in the reference dataset. These additional road segments were considered as false positives by the accuracy metrics as they were not present in the reference data. Previous studies have shown that inaccuracies in the reference dataset can underestimate the accuracy (e.g., ref. [37]). We did not attempt to recalculate the metrics using improved reference data, because the U-Net model's qualitative visual interpretation of road detection pointed to a satisfactory result.

### 3.3. Estimation of Road Extension in the Brazilian Legal Amazon

The following step in our analysis was to test the U-Net road model's performance on a large-scale prediction of roads for the entire Brazilian Legal Amazon territory (approximately 5 million square kilometers). First, we created a mosaic composed of the Sentinel-2 RGB image color composite from 2020 with the same parameters as the images used to train, test, and validate our model. The road prediction and post-processing ran in the Azure planetary computer platform for approximately seven hours to return the predicted roads and clean the data, reducing its false positives, connecting loose segments, and converting images to vectors for better geospatial analysis. In addition, we managed to speed up the computational process by utilizing the in-memory file system of Gdal (*vsimem*), which saved each data information on memory instead of holding each 256 × 256 pixel input and output image on disk.

The U-Net road model identified 3.46 million kilometers of roads covering the Legal Amazon area, spreading across nine states. The mapping area of this study was larger than

the ARD one, which focused mainly on the forest biome. We split our analysis of road extension into two perspectives: a general and per land category. The general approach calculated the roads' presence throughout the Legal Amazon area by estimating the road density (Figure 9). The density map used a 10 km × 10 km cell grid (i.e., 100 square kilometers) to estimate the length of roads present in each cell. At the same time, the histogram calculated the road density distribution (Figure 10). We found that the density varied between 0 and 5.75 km·km$^{-2}$, mainly concentrated around the south, southeast, and east regions of the Legal Amazon (Figure 9). Additionally, the central and northwest regions have fewer road presences; therefore, the calculated density was approximately near-zero or inexistent.

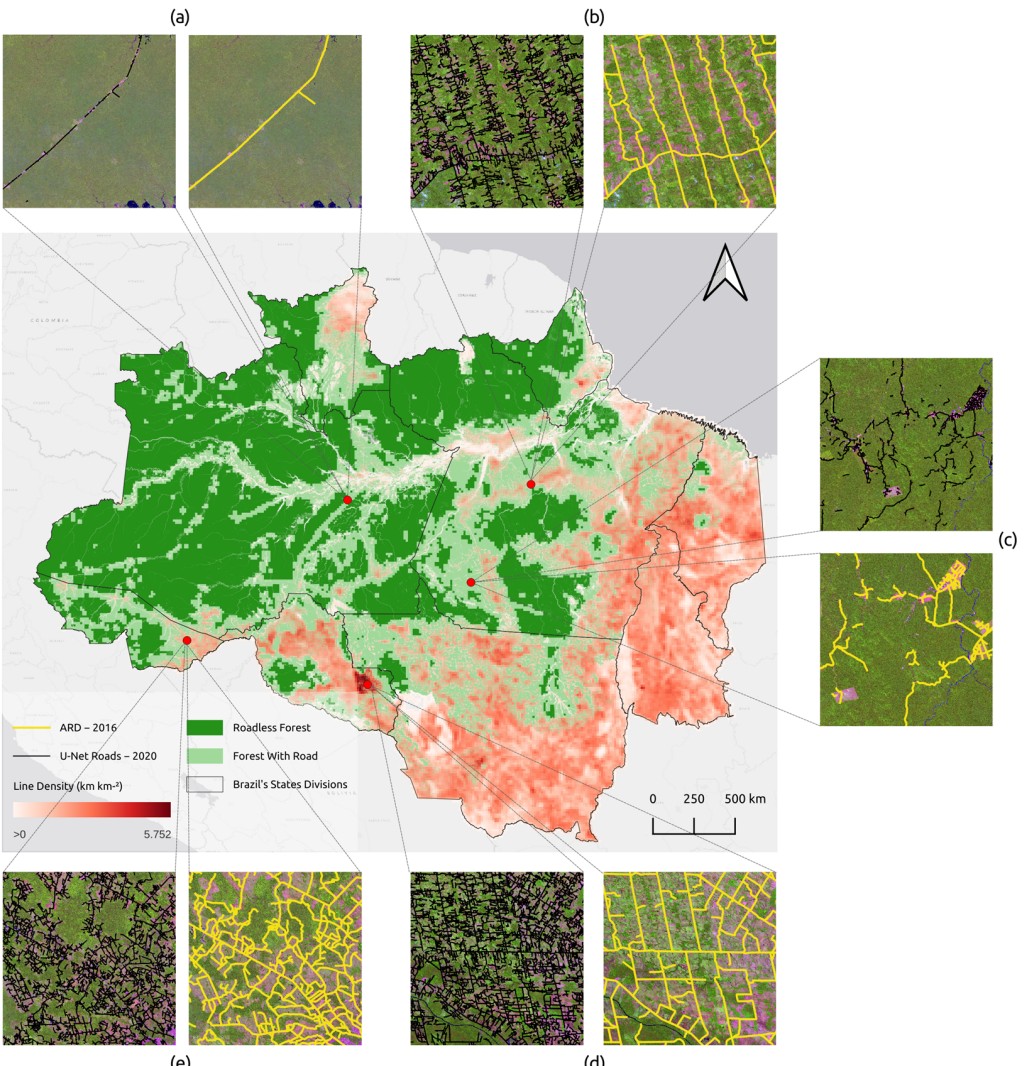

**Figure 9.** Road density obtained with the U-Net road model. A comparison of the U-Net model (2020) and the Amazon Road Dataset (ARD, 2016) is shown in the image panels: (**a**) the BR-319 highway with unconnected segments by the U-Net road model (black line) and ARD showing the full connected length; (**b**) the fishbone road pattern of the Trans-Amazonia highway main road (BR-230) and perpendicular ones; (**c**) a typical road pattern of selective logging; (**d**,**e**) a geometric road pattern in agriculture lands.

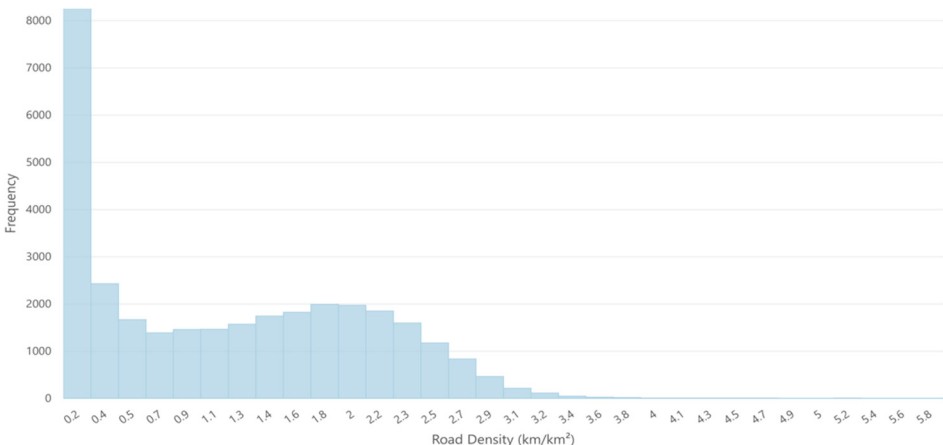

**Figure 10.** Histogram of road density detected with the U-Net road model within a 10 km × 10 km cell throughout the Brazilian Legal Amazon.

The road density map's histogram reinforced this information. According to the distribution, a higher number of grid cells in the road density map ranges from 0 to 0.2 km·km$^{-2}$. This lower road density region concentrates in areas mostly covered by forest and along the major rivers (Figure 9). The lower road density zone extends until 0.5 km·km$^{-2}$ on the transition to the so-called Arc-of-Deforestation, a region where land occupation is consolidated. The road density in the consolidated frontier ranges from 0.5 km·km$^{-2}$ to 3.2 km·km$^{-2}$. There are a few hotspots where the road density is higher than 3.2 km·km$^{-2}$ in rural settlements and peri-urban regions (Figures 9 and 10).

Furthermore, we assessed the road extent and density by state (Table 1) and land category (Table 2). We calculated each state's road length and average density within the Brazilian Legal Amazon. We observed that the state of Tocantins had the highest average road density of 1.77 km·km$^{-2}$ comprising 490 thousand kilometers of road, followed by Maranhão, Mato-Grosso, and Rondônia, each having a density of 1.57, 1.44, and 1.31 km·km$^{-2}$, respectively (Table 1). These four states are the only ones that surpass the 1 km·km$^{-2}$ average road density mark, indicating a more continuous construction of road networks in their territory. The two largest states, Pará and Amazona, showed similar average road densities (0.57 and 0.56 km·km$^{-2}$, respectively), but the road extent in Pará was nine times higher than that in Amazonas (Table 1). Mato Grosso had the most considerable extent of roads with 1.3 million km, followed by Pará with 45% less (i.e., 715 thousand km) of roads in that state. Acre and Roraima had a similar road density and Amapá had the lowest one (Table 1).

**Table 1.** Comparison between the length of roads detected per state in the Legal Amazon Area.

| States | Road Length (km) | Average Road Density (km·km$^{-2}$) |
|---|---|---|
| Acre | 53,614 | 0.33 |
| Amazonas | 79,801 | 0.56 |
| Amapá | 25,010 | 0.02 |
| Maranhão | 412,306 | 1.57 |
| Mato Grosso | 1,296,946 | 1.44 |
| Pará | 715,730 | 0.57 |
| Rondônia | 310,119 | 1.31 |
| Roraima | 80,025 | 0.36 |
| Tocantins | 490,513 | 1.77 |
| Total | 3,464,066 | |

**Table 2.** Road extent and density by land category.

| Land Category | Data Source | Road Extent (km) | Relative Extent (%) | Average Road Density (km·km$^{-2}$) |
|---|---|---|---|---|
| Indigenous Land | ISA | 91,579 | 2.6% | 0.08 |
| Federal Protected Area | ISA | 42,319 | 1.2% | 0.07 |
| State Protected Area | ISA | 141,735 | 4.1% | 0.28 |
| Quilombo | INCRA | 4904 | 0.1% | 0.82 |
| Military Area | CNFP | 1713 | 0.0% | 0.07 |
| Rural Settlement | INCRA | 426,139 | 12.3% | 1.27 |
| Public Forest | CNFP | 7660 | 0.2% | 0.10 |
| Private Land | INCRA | 1,893,738 | 54.7% | 1.46 |
| Public Land | Imazon | 854,279 | 24.7% | 0.72 |
| Total | | 3,464,066 | | |

Understanding road expansion in a land category is also necessary to assess its impact on deforestation and logging, road pressure in protected areas, and the appropriation of public lands (Table 2). Most roads concentrate on private land with 1.9 million km (i.e., 54%), with a road density of 1.46 km·km$^{-2}$ (Table 2). Astonishingly, 24.7% of the total road extent mapped with U-Net, i.e., 854 thousand km, is within public land, which is open and vulnerable to illegal appropriation, a process known as land grabbing. The third-largest occurrence of roads is within rural settlements with 426 thousand km (12% of the total extent), where the fishbone deforestation pattern prevails. Protected areas for conservation and sustainable use combined, including Indigenous Land, Federal and State Protected areas, and Quilombo, concentrate 8% of the total road mapped (i.e., 280.5 thousand km; Table 2). The most threatened categories of protected areas are State (4.1%) and Indigenous Land (2.6%).

## 4. Discussion

With the development of new cloud-based computing technologies and advances in Artificial Intelligence, large-scale mapping for various scenarios has become faster and more common [30,38,39]. In addition, open-source code allowed for an adaptation of the original U-Net architecture to train and map roads in the Amazon in a fraction of the time required to identify and trace roads [8] visually. Our new U-Net road model detected approximately 3.46 million kilometers of roads in the Legal Amazon region within 7 h of cloud computing to run the U-Net model and post-processing and vectorize the data. This process generally would take several months of human analysis and visual interpretation. The previous effort to monitor road expansion in the Amazon region paid off because it provided valuable data for assessing the negative road impact on the Amazon. The ARD also offered vital information to build the new AI model by statistically selecting areas to create the training, calibration, and road datasets. The new Brazilian Amazon road map for 2020 cannot be directly compared with the ARD from 2016. First, the latter focused the road mapping in the Amazon biome region, an area 22% smaller than the Brazilian Legal Amazon. Second, the U-Net road map is more detailed than the ARD one because it detects more road segments, especially in rural properties.

We estimated the forest area affected by roads in the Amazon biome with the road density map within a 10 km grid cell. We used the forest area obtained from the MapBiomas project [40] for 2016 and 2020 to estimate the expansion of roads in forested areas (excluding 0.2 million km$^2$ of second-growth forests and highly degraded forests). We found that of the 3.1 million km$^2$ of remaining forest, 1.83 million km$^2$ were roadless forests (i.e., 59%). The remaining forests in 2020, i.e., 1.27 million km$^2$ (41%), are carved by roads or within 10 km of all roads (Figure 9) and, consequently, are under deforestation and forest degradation pressure. Further analysis is required to understand the relationship of roads with forest fragmentation, fires, and deforestation in light of this new road dataset we obtained with

AI U-Net road post-AI detection models, as well as the estimation of the roadless forests in the Brazilian Amazon.

Several applications can also be implemented with the new road dataset obtained with the AI U-NET mode. Previous analysis of the correlation of road distance with deforestation [41] and fires [6] can be updated. We already used the new AI-mapped road to update a deforestation risk model for the Amazon. As one of the predictor variables, the original deforestation risk model used the ARD map based on visual interpretation with data from 2016 [16]. The deforestation risk model depends on the annual update of roads (as the distance to roads is one of the most important predictor variables [16], a task considered prohibitive using human interpretation). The new AI road model allowed us to update the ARD to 2020 and operationalized the deforestation risk model (see: https://previsia.org accessed on 1 June 2020).

Other applications can be explored with the new AI ARD, such as transportation and logistic planning for agriculture and agroforestry commodities, model landscape fragmentation, determining roadless forest landscape, and access to logging and settlement roads, e.g., Ref. [13]. The new AI road detection model will allow for keeping the ARD updated for these and other applications and support forest conservation efforts and the protection of (open) public lands.

Although our results are accurate and somehow better than the road maps produced with human interpretation of satellite imagery, some challenges persist in reducing false-positive road detection. One possible and standard solution to overcome this problem is to improve the training dataset of roads by increasing data samples and data variation. This is the case for our model because most of the false positives were found in areas with a low density of training data. The second-largest mapping issue is the discontinuity of road segments, which happens for various reasons. First, the spectral signal of roads can be obscured by vegetation cover and by the spectral similarity of adjacent pasture and agricultural lands with the road substrate, which is mainly formed by dirty compacted soil. The road discontinuities can be partially fixed by automatically setting line snapping thresholds, but this works only for small distances (i.e., <100 m). Further research and AI modeling is needed to determine a fully automatic solution to deal with fragmented roads; meanwhile, a task will be implemented with human aid. We also recommend comparing our proposed U-Net road detection model with another AI algorithm for future research. Our next goal is to expand our model to create historical maps of roads in the Amazon, with available Landsat and Sentinel datasets. Preliminary learning transfer tests applying our U-Net AI road model built with Sentinel-2 to Landsat imagery showed promising results.

## 5. Conclusions

The ARD, generated with human effort throughout the years, together with open-source algorithms and cloud-based computing, allowed the development of a new AI road detection model. The updated road maps for the Amazon region will enable the implementation of scientific, societal, and policy applications. The correlation of roads with deforestation, fire occurrence, and landscape fragmentation can be further investigated with the more detailed and extensive AI road dataset. Our initial learning transfer of the AI road model obtained with Sentinel-2 imagery is promising to apply the U-Net road model to the Landsat data archive to reconstruct the road dynamic of the region. Further research includes improving the post-road center line vectorization of fragmented roads and road categorization. Finally, our results pointed out that large portions of the Brazilian Amazon forests are dominated by an intricate and complex road network expanded from the main official roads and prolonged over the pristine forests and protected areas for conservation. The AI U-Net road model revealed more roads than the human-based road mapping efforts requiring further research to understand their negative ecological impacts. In addition, because roads predominantly come first, the new AI road detection model opens up an untried class of forest monitoring, avoiding future deforestation and forest degradation.

**Author Contributions:** Conceptualization, C.M.S.J. and J.B.J.; methodology, C.M.S.J. and J.B.J.; software, J.B.J. and S.C.P.C.; validation, J.B.J. and J.G.R.; formal analysis, C.M.S.J. and J.B.J.; investigation, C.M.S.J. and J.B.J.; resources, J.G.R. and S.C.P.C.; data curation, J.B.J. and J.G.R.; writing—original draft preparation, J.B.J. and C.M.S.J.; writing—review and editing, J.B.J. and C.M.S.J.; visualization, J.B.J. and S.C.P.C.; supervision, C.M.S.J.; project administration, C.M.S.J.; funding acquisition, C.M.S.J. All authors have read and agreed to the published version of the manuscript.

**Funding:** This research received funding from the Climate Land Use Alliance (CLUA, Grant Number G-2008-57051), Fundo Vale (#009/2020), and cloud computing credits and technology development from Microsoft AI for Earth to support the Imazon research team, which we are profoundly thankful.

**Data Availability Statement:** The U-Net road map is available at https://previsia.org/ (accessed on 1 May 2022). Please, contact the corresponding author for data access. The U-Net road model and its Python code are available https://github.com/JonasImazon/Road_detection_model.git (accessed on 1 May 2022).

**Acknowledgments:** We acknowledged and are thankful for the technical support from Microsoft and Radix to implement our U-Net road model in Azure cloud computing. We also thank Jailson Souza Filho, Imazon's data scientist, to support the analysis presented in Figure 9.

**Conflicts of Interest:** The authors declare no conflict of interest.

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
