# Peer review of "Mapping Roads in the Brazilian Amazon with Artificial Intelligence and Sentinel-2"

_remotesensing, doi:10.3390/rs14153625_

Round 1

Author Response

Reviewer 1                                                                              

The authors introduce an automated detection method to map roads in the Brazilian Amazon using Sentinel 2 images. They used a modified U-Net model and a post-AI detection protocol. Some issues need to be addressed.                                       

{Author's Response}

Thanks for your review and suggestions to improve our manuscript. Please, find below the answers to your questions.

  1. The title talks about mapping roads by Artificial Intelligence. But, this paper focuses on mapping roads by a modified U-Net and does not involve other Artificial Intelligence technologies. Please modify the title to make it specific.

{Author's Response} Thanks for suggesting making the title more specific. We have thought about it and concluded that a more general title would potentially reach out to more readers. Several research papers use 'Artificial Intelligence' but use just one technology in this large field, as indicated in the link below

https://scholar.google.com.br/scholar?q=mapping+roads+with+artificial+intelligence&hl=pt-BR&as_sdt=0&as_vis=1&oi=scholart

Therefore, we respectfully decided to keep the title as it is.

  1. Line 22, “We found and astonishing foot print of roads in the Brazilian Legal Amazon, with 3.4 million km 22 of roads mapped in 2020.” Please check this sentence.

{Author's Response} Fixed! We deleted 'd' to read "We found an astonishing footprint…', and the word 'footprint' was also fixed (blank space deleted).

  1.                                                                                   
    Line 25, km2→km                                                                                                                                                       
  2. The introduction section does not provide a succinct theoretical basis for the study. Please expand and highlight the advantages of the modified U-Net model.

{Author's Response} We have expanded the third paragraph of the introduction with a more theoretical basis of the U-Net (lines 67-84), as indicated below:

Automated road detection methods have been developed for very high spatial resolution imagery based on deep learning algorithms [12,20]. Artificial Intelligence (AI), particularly convolutional neural network (CNN) deep learning algorithm, has been used successfully to detect and map rural roads in large forested areas of Canada with RapidEye imagery [11]. The AI-CNN method in Canada did not produce full-connected vectorized roads, requiring post-classification techniques, resulting in recall accuracy of 89–97% and precision of 85–91% [11]. A U-Net [21], a CNN variant, was first used for segmentation in medical imaging. It is [22] considered a state-of-art for image segmentation because it is based on the deconstruction and reconstruction of images for feature extraction, increasing object detection in various applications [2,23,24]. U-Net's differentiating characteristics are its accuracy and high-speed discriminative learning capacity from a small number of training images [21]. As a result, Several U-Net remote sensing applications have been proposed, including road extraction using high-resolution imagery [2,22–24]. Therefore, the U-Net algorithm is promising to automate the detection of rural roads of the Brazilian Amazon, overcoming the visual interpretation mapping protocol [8] used to build the ARD. We modified the original U-Net algorithm, which erodes image input chips (256 x 256 pixels), making road discontinuity a problem, and changed the activation and loss functions of the model to make it more sensitive to detect roads using Sentinel 2 10-m spatial resolution imagery.

  1. The main goal of this paper is to develop an automated AI model to monitor
    roads more frequently in the Brazilian Amazon and assess the road impact on deforestation. (Line 86-88) However, the assessment of the road impact on deforestation was not presented in the Results or Discussion.

{Author's Response} We appreciate you bringing that up. In fact, the word 'deforestation' is a mistake. Our study assesses the impact of roads on forests. We fixed and it reads as follows:

Our main goal with this study is to develop an automated AI model to monitor roads more frequently in the Brazilian Amazon and assess the correlation of forests with roads.

  1. The words in Figure 1 are difficult to discern.

{Author's Response} We have increased the font size, improved the figure resolution, fixed typo problems, and made it more concise.

7. Please add a scale in Figure 2.

{Author's Response} Done.                                                                                                                                                                                  
8. Line 193, the samples for training, validation, and testing cannot be separated in Figure 1. And how the training dataset was chosen?

{Author's Response} Figure 1 was fixed, indicating the three types of sample datasets.  The sub-section 2.4. (Sampling and Data Preparation) explains how these datasets were obtained. First, we subset the existing dataset (Amazon Road Dataset) and clipped it with a grid of 256x256 cells, resulting 2500 sample chips, which were subsequently split into training calibration and test datasets.                                                                                                                                                                                              
9. Line 230, the link cannot be accessed.

{Author's Response} Sorry, the access was not public. We fixed it; it is possible to access the link now.                                              

  1. 30% of the 2500 chip samples are separated equally for validation (375 chips). and testing (375 chips) (Line 193). However, the accuracy is only computed by validation (250 chips) and test (250 chips) datasets (Line 232-233). How the validation (250 chips) and test (250 chips) images were chosen?

{Author's Response} The splitting was 80% for training, 10% for calibration, and 10% for testing. We randomly split these datasets from the initial 2500 sample chips, and the text reads as follows:

We randomly selected 80% of our data for training purposes and separated the 20% equally for validation and test, i.e., 250 each (Figure 1).

  1.                                                                        
    Line 260, why the threshold of 20% was selected?

{Author's Response} The output of the U-Net model is a probability image of a pixel classified as a road. The 20% threshold was defined empirically to highlight pixels most likely to be a road. We added this information to the manuscript to read as:

To detect the road center line, we first empirically define a confidence threshold of 20% (i.e., pixels with a probability value equal to or greater than 0.2) over the output values, creating a road binary image.       

  1. Line 282-286, please explain the advantages of these four steps.

{Author's Response} We added the explanation below to the manuscript:

These steps allowed us to deal with different categories of false positive road detection in an automated manner.

  1. Please modify the typo in Figure 7.

{Author's Response} We fixed the typo problems of Figure 7.                                               

  1. Please clearly presents your visual results in Figure 7.

{Author’s Response} We have made Figure 7 by changing the name of columns (d) and (e) to ‘Prediction’.

  1. Line 364, 3.4 → 3.3

{Author's Response} We changed 3.4 on line 364 to 3.3.

  1. Line 378, Figure S1 cannot be accessed.

{Author's Response} Thanks for pointing it out. We deleted Figure S1 because there isn’t supplemental material. We have thought about including it but decided not to.

  1. (a)-(e) cannot be found in Figure 9.

{Author's Response} We fixed Figure 9 caption by adding the indicators of images (a)-(e).

  1. Line 406, “densityh” ?

{Author's Response} We fixed “densityh” to “density”.

  1. The controlled experiments lacked in this paper.

{Author's Response} We followed the well-established remote sensing statistical protocol to control the experiment. First, we designed an unbiased statistical random sampling. Then, we randomly selected the training, validation, and test samples. We used test samples to assess the U-Net road detection following traditional deep learning result assessment. This protocol is presented in the manuscript.        

Reviewer 2 Report

The author proposed a U-Net based method to detect roads in the Brazilian Amazon area, I have some comments:

1. The resolution of Figure 1 is too low, please redraw this workflow.

2. There are a lot of road mapping or road extraction proposed recently. However, the authors only referred to [12, 20], I suggest that authors should refer to newer related works.

3. Specific and clear contributions should be listed in section 1 - Introduction. The most important is that there are a lot of road mapping and road extraction proposed recently. What are the advantages of the proposed method in this manuscript? 

4. Ablation experiments should be conducted to compare U-Net road model with original U-Net model.

Author Response

Reviewer 2

Comments and Suggestions for Authors

The author proposed a U-Net based method to detect roads in the Brazilian Amazon area, I have some comments:

  1. The resolution of Figure 1 is too low, please redraw this workflow.

{Author's Response} We have increased the font size and improved the figure resolution, and fixed typo problems.

  1. There are a lot of road mapping or road extraction proposed recently. However, the authors only referred to [12, 20], I suggest that authors should refer to newer related works.

{Author's Response} In addition to [12, 20], we had also cited:

  1. Stewart, C.; Lazzarini, M.; Luna, A.; Albani, S. Deep Learning with Open Data for Desert Road Mapping. Remote Sensing 2020, 12, doi:10.3390/RS12142274.
  2. Gao, L.; Song, W.; Dai, J.; Chen, Y. Road Extraction from High-Resolution Remote Sensing Imagery Using Refined Deep Residual Convolutional Neural Network. Remote Sensing 2019, 11, 1–16, doi:10.3390/rs11050552.
  3. Augustauskas, R.; Lipnickas, A. Improved Pixel-Level Pavement-Defect Segmentation Using a Deep Autoencoder. Sensors 2020, Vol. 20, Page 2557 2020, 20, 2557, doi:10.3390/S20092557.

  1. Specific and clear contributions should be listed in section 1 - Introduction. The most important is that there are a lot of road mapping and road extraction proposed recently. What are the advantages of the proposed method in this manuscript?

{Author's Response} We added sentences highlighting the significance of our study and the U-Net model to detect roads using Sentinel 2 moderate spatial resolution imagery. Most road detection studies use fine resolution imagery.

  1. Ablation experiments should be conducted to compare U-Net road model with original U-Net model.

{Author's Response} The main difference between the original and the new U-Net road model is the reconstruction of the edge of the output image. In addition, we also included a modification on the loss function, replacing the original cross entropy loss function with a Soft Dice Loss Function to maximize the matching of pixels between the output and the reference data. Another modification was the usage of the Leaky ReLu activation function to prevent the dying neuron problem caused by the ReLu function, which due to the high imbalance of our dataset, caused most of the network’s neurons to have low or zero influence on the model’s outcome. The original U-Net erodes the input image creating a gap of information along the edge, which increases road discontinuity. Because of that, it is unnecessary to conduct an ablation experiment to compare the U-Net models. The original U-Net had to be modified to eliminate the edge erosion problem.

Reviewer 3 Report

In this paper, the authors apply the U-Net and several post-preprocessing algorithms for road segmentation in the amazon area. The work is interesting and meaningful to the amazon forest area. However, the entire work still needs several improvements before the final publication. The followings are my main comments for this paper.

1.  The 3-band input image for U-NET includes red, NIR,and SWIRI and the authors state these channels are more suitable for detecting roads in non-urban areas. Can authors add more explanation about it? Why these channels are better than the green or blue channels? Maybe the authors can just show a similar image in figure 3(b) of the green/blue channel for comparison.

In addition, in the entire manuscript the author call red, NIR, and SWIRI image as RGB image (such as the title of figure 3). I assume only a red-green-blue image can be called RGB. Please fix the name of it.

2. In section 2.4, the authors say they use data augmentation to solve the underfitting problem. Actually, increasing the training data size can solve the overfitting problem instead of underfitting. Please correct it.   As for the data augmentation, the authors apply the three repetitions for each image, and how authors split the whole data into training, validation, and test sets? Are three same images in the same set or the different sets?  

3. The authors mention the hyperparameter of U-Net in section 2.5 and how do you do the hyperparameter tuning process? What methods do you use? The author mention they use early stopping to avoid overfitting and please also list the parameters for it. 

4. The input image size is 256*256*3 and how authors select this dimension? Do you compare several different sizes of images and then find that 256*256*3 performs best? 

5. In section 3.1, the authors mention they capped epoch iterations at the 25th epoch, does it mean the training process ends at the 25th epoch due to early stopping? The authors mention the 17th epochs get the smallest loss, so do all prediction results shown in figure 7 and figure 8 are generated based on the 17th epoch? 

6. In figure 8, the validation and test accuracy are better than the training dataset, which is wired, and can you explain why this happens? Typically, when your training and test datasets are in different distributions can cause this problem. In addition, you can also check and modify the regularization of your U-Net model to avoid this problem.

7. I don't find figure S1 in this manuscript.

8. In figure 9, please add the (a), (b), (c), (d), and (e).

9. In the results section, there is no comparison between the proposed U-net model and other deep learning models like DenseNet or SegNet. It is necessary to compare your model with other SOTA methods to verify the superiority of your model in road segmentation. It will be reliable that the authors add at least two more comparison methods in the results section. 

Author Response

Reviewer 3

Comments and Suggestions for Authors

In this paper, the authors apply the U-Net and several post-preprocessing algorithms for road segmentation in the amazon area. The work is interesting and meaningful to the amazon forest area. However, the entire work still needs several improvements before the final publication. The followings are my main comments for this paper.

{Author's Response} Thanks for the in-depth review of our manuscript. We addressed all the issues identified and most suggestions.

  1. The 3-band input image for U-NET includes red, NIR,and SWIRI and the authors state these channels are more suitable for detecting roads in non-urban areas. Can authors add more explanation about it? Why these channels are better than the green or blue channels? Maybe the authors can just show a similar image in figure 3(b) of the green/blue channel for comparison.

{Author's Response} We have conducted previous studies on mapping roads with Landsat imagery in the Amazon through visual interpretation and hand digitizing roads (references below). Most roads in rural areas of the Amazon region are dirty and unpaved, with soil exposed and high surface reflectance values in the SWIR, NIR, and Red spectral bands. The Blue and the Green spectral bands have lower soil reflectance, making road detection a challenge. We assigned the SWIR, NIR, and Red spectral bands (not channels!) in the RED=SWIR, GREEN=NIR, and BLUE=Red color channels to create a color composite (the colored image in Figure 3) that enhances road detection. We did not add  the Green and Blue spectral bands, as suggested because  it would make the figure 'polluted' with unnecessary information; we kept the original Figure 3. We noticed that the order of the band SWIR1 and Red were flipped and then fixed to their right color channels to indicate the right color composite used in the U-Net model.

We also have indicated in the manuscript that "... the bands SWIR1, NIR, and Red are more suitable for detecting roads in non-urban areas and differentiating from other linear features (e.g., powerlines, and geological lineaments, among others) [8].

Reference about road spectral characteristics in Amazonian forests cited in the manuscript:

Brandão, A.O.; Souza, C.M. Mapping Unofficial Roads with Landsat Images: A New Tool to Improve the Monitoring of the Brazilian Amazon Rainforest. International Journal of Remote Sensing 2006, 27, 177–189, doi:10.1080/01431160500353841.

Nascimento, E. de S.; da Silva, S.S.; Bordignon, L.; de Melo, A.W.F.; Brandão, A.; Souza, C.M.; Silva Junior, C.H.L. Roads in the Southwestern Amazon, State of Acre, between 2007 and 2019. Land (Basel) 2021, 10, 1–12, doi:10.3390/land10020106.

In addition, in the entire manuscript the author call red, NIR, and SWIRI image as RGB image (such as the title of figure 3). I assume only a red-green-blue image can be called RGB. Please fix the name of it.

{Author's Response} As stated above, RGB channels are the primary color channels that can receive a variate of spectral bands to derived color composites (see examples here: https://custom-scripts.sentinel-hub.com/custom-scripts/sentinel-2/composites/). We assigned the SWIR, NIR, and Red spectral bands to these color channels: RED=SWIR, GREEN=NIR, and BLUE=Red. This procedure creates a false-color image (a true color composite would be RED=Red, GREEN=Green, BLUE=Blue). We kept the name in the original manuscript because it is correct.

  1. In section 2.4, the authors say they use data augmentation to solve the underfitting problem. Actually, increasing the training data size can solve the overfitting problem instead of underfitting. Please correct it. As for the data augmentation, the authors apply the three repetitions for each image, and how authors split the whole data into training, validation, and test sets? Are three same images in the same set or the different sets?

{Author's Response} As suggested, we correct it by replacing 'underfitting' with 'overfitting.' The splitting was 80% for training, 10% for calibration, and 10% for testing. We randomly split these datasets from the initial 2500 sample chips and proceeded with the data augmentation process. The replicated images remain in the same set because the data splitting comes first.

  1. The authors mention the hyperparameter of U-Net in section 2.5 and how do you do the hyperparameter tuning process? What methods do you use? The author mention they use early stopping to avoid overfitting and please also list the parameters for it.

The hyperparameter tuning process was done based on trial-and-error attempts with multiple different values for learning rate, optimizers, LeakyRelu alpha value, loss functions, number of layers, dropout rate, and early stopping. Since our study objective is not to assess the effect of hyperparameters on the model, we only focused on obtaining the best model result through the training and calibration process described in the manuscript. We configure the early stopping to have the patience of 3 epochs over the improvement of the validation loss value while also restoring the neural connections’ weights based on the best epoch with the lowest validation loss (17th epoch).

  1. The input image size is 256*256*3 and how authors select this dimension? Do you compare several different sizes of images and then find that 256*256*3 performs best?

{Author's Response} The U-Net model can handle two image sizes: 512 x 512 and 256x256. This dimension was selected because we want to capture road textural and spectral characteristics at the local level. Adding a larger image chip would smooth the pixel characteristics of roads. We previously compared the different sizes during the model calibration and opted for the model with the smaller one because it better detects roads. We did not include this in the analysis because the calibration test was conducted in smaller areas.

  1. In section 3.1, the authors mention they capped epoch iterations at the 25th epoch, does it mean the training process ends at the 25th epoch due to early stopping? The authors mention the 17th epochs get the smallest loss, so do all prediction results shown in figure 7 and figure 8 are generated based on the 17th epoch?

{{Author's Response} We capped the number of iterations at the 25th epoch because, during the multiple training/calibration iterations, the loss value stabilized between the 21st and the 23rd iterations, so we rounded up. And yes, the predictions shown in Figures 7 and 8 were generated based on the 17th epoch, which got the smallest validation loss of all iterations. We added a sentence explaining the setting of the 25th epoch to the manuscript, as follows:

 “...The 25th epoch mark was set due to the early stopping results during the calibration phase.”

  1. In figure 8, the validation and test accuracy are better than the training dataset, which is wired, and can you explain why this happens? Typically, when your training and test datasets are in different distributions can cause this problem. In addition, you can also check and modify the regularization of your U-Net model to avoid this problem.

{Author's Response} The validation and test accuracy seldom surpass the training accuracy as you pointed out. Our sampling design protocol randomly selected the training, validation, and test data sets to ensure that the probability distributions in these sets closely resemble those in the training set. Since there is usually some overfitting, which lowers validation and test accuracy, the training accuracy is often higher (mainly if you run enough epochs, as I see you did). We did not run enough epochs because the loss value stabilized soon, but this may not explain higher test accuracy relative to the training one. We also added a dropout layer, which is the only other factor that can explain higher test accuracy. The difference between the accuracies can be explained based on the usage of Dropout as a regularization method during the training phase. The dropout randomly deactivated 30% percent of the model’s final layer to reduce the possibility of overfitting. It is known that this method can reduce the accuracy of the training while increasing the generalization of the validation and test datasets. We have not explored this in the manuscript because it needs further analysis, which can be the subject of a more specialized AI model study.

  1. I don't find figure S1 in this manuscript.

{Author's Response} Thanks for pointing it out. We deleted Figure S1 because there is no supplemental material.

  1. In figure 9, please add the (a), (b), (c), (d), and (e).

{Author's Response} Fixed. Thanks for picking this up.

  1. In the results section, there is no comparison between the proposed U-net model and other deep learning models like DenseNet or SegNet. It is necessary to compare your model with other SOTA methods to verify the superiority of your model in road segmentation. It will be reliable that the authors add at least two more comparison methods in the results section.

{Author's Response} Comparing the U-Net model with other ones is not the objective of this manuscript. Model comparison is not required since many research articles apply a single AI model for solving a problem.

Round 2

Reviewer 1 Report

The authors have addressed most of my questions in the revised manuscript. Nevertheless, a comparison of the results by the proposed method and other deep learning methods should be added to their manuscript if they want it to be considered for publication.

Author Response

Reviewer 1

Comments and Suggestions for Authors

The authors have addressed most of my questions in the revised manuscript. Nevertheless, a comparison of the results by the proposed method and other deep learning methods should be added to their manuscript if they want it to be considered for publication.

{Authors' Answer} Thanks for your valuable comments and suggestions on the previous version of our manuscript. We respectfully disagree with the remaining recommendations of adding 'a comparison of the results by the proposed method and other deep learning methods.' First, comparing AI methods is not the objective of our study. We have searched the literature for the most promising methods and chose the U-Net model because of its potential robustness in detecting roads. Secondly, it is not mandatory for a scientific publication to include a comparison of results with different methods (unless it is the study's primary objective). There are several research articles on AI solutions, with only one method published. Finally, research projects, as you may know, are conducted in cycles. We completed the first cycle successfully. We added the following statement suggesting future research as you proposed in the discussion section: 

"We also recommend comparing our proposed U-Net road detection model with another AI algorithm for future research.".

Reviewer 2 Report

After careful reviewing, I think the authors have addressed all my questions, and I have no other extended questions.

Author Response

Reviewer 2

Comments and Suggestions for Authors

After careful reviewing, I think the authors have addressed all my questions, and I have no other extended questions.

{Authors' Answer} Thanks for your valuable comments and suggestions on the previous version of our manuscript.

Reviewer 3 Report

In this revised version of the manuscript, the authors address all my comments and I think the paper can be published.

Author Response

Reviewer 3

Comments and Suggestions for Authors

In this revised version of the manuscript, the authors address all my comments and I think the paper can be published.

{Authors' Answer}  Thanks for your valuable comments and suggestions on the previous version of our manuscript.